# Engineering of Nisin as a Means for Improvement of Its Pharmacological Properties: A Review

**DOI:** 10.3390/ph16081058

**Published:** 2023-07-26

**Authors:** Mateusz Musiejuk, Paweł Kafarski

**Affiliations:** Faculty of Agriculture and Forestry, University of Warmia and Mazury, pl. Łódzki 4, 10-957 Olsztyn, Poland; mateusz.musiejuk@uwm.edu.pl

**Keywords:** antibiotic resistance, nisin, lantibiotics, protein engineering, bioengineering

## Abstract

Lantibiotics are believed to have a conceivable potential to be used as therapeutics, especially against clinically resistant bacterial strains. However, their low solubility and poor stability under physiological conditions limit their availability for clinical studies and further pharmaceutical commercialization. Nisin is a readily available and cheap lanthipeptide and thus serves as a good model in the search for the tools to engineer lantibiotics with improved pharmacological properties. This review aims to address technologies that can be applied to alter and enhance the antimicrobial activity, antibacterial spectrum and physicochemical properties (solubility, solution stability and protease resistance) of nisin. There are basically two general means to obtain nisin analogs—protein engineering and chemical functionalization of this antibiotic. Although bioengineering techniques have been well developed and enable the creation of nisin mutants of variable structures and properties, they are lacking spectacular effects so far. Chemical modifications of nisin based on utilization of the reactivity of its free amino and carboxylic moieties, as well as reactivity of the double bonds of its dehydroamino acids, are in their infancy.

## 1. Introduction

Bacterial infections are among the most important diseases and they still account for a substantial proportion of deaths worldwide. For decades, multiple varieties of antibiotics have not only been used for therapeutic purposes but applied prophylactically in human medicine and across agriculture and animal husbandry [1,2,3]. This has led to the substantial emergence of antibiotic resistance against commonly used frontline treatments and this resistance represents an alarming sign for both human and animal healthcare. Moreover, new mechanisms of resistance continue to emerge and spread globally [3,4,5]. Thus, despite the fact that the development of new antibiotics has slowed dramatically since the 1950s’ golden age of discovery, there is still a considerable interest of academic and industrial laboratories for the delivery of new anti-microbial drug classes. It is, however, not an easy task because the cost of bringing a new antibiotic from discovery to market is high and return on investment is relatively low [6,7]. 

Among strategies to tackle antibiotic resistance that have been under investigation, are the use of prophylactic vaccines, therapeutic monoclonal antibodies, phage therapy and phage enzymes, the use of plasmid curing agents, quorum sensing inhibitors, nanomedicines, drug repurposing and, finally, antimicrobial peptides [8,9,10].

In the latter respect promising candidates appeared to be peptides produced by bacteria and targeting peptidoglycans in pathogenic bacterial cell walls. Due to this mechanism of action, they usually show little or no cross-resistance and a lack of harm to humans [11,12]. Among them, ribosomally synthesized ones constitute an emerging class of natural products that have attracted considerable interest. Within this diverse group of peptides, the lantibiotics, called class I bacteriocins—small proteins produced by lactic acid bacteria—have become the focus of many biomedical and pharmaceutical research groups [13,14]. These bacteriocins are distinctly different from conventional antibiotics in their biosynthesis, multiple modes of action and demonstrable high potency in vitro (also towards highly resistant strains), and ability to destroy target cells rapidly. 

Despite these promising attributes, there are also a number of limitations that have prevented their more widespread use so far. These include poor pharmacokinetics, instability and/or insolubility at physiological pH, as well as some susceptibility to proteolytic digestion [15,16]. To overcome this limitation, the implementation of multiple technologies to obtain modified lantibiotic structures have been utilized. Various approaches of protein engineering (both in vivo and in vitro) and a search for novel naturally occurring variants of lantibiotics have been applied. Their chemical modifications have attracted far less interest and are quite scarce. 

This review outlines some of the more recent developments on the implementation of in vivo and in vitro engineering technologies to alter, and even enhance, the antimicrobial activity, antibacterial spectrum, solubility, stability and protease resistance of nisin. These studies are designed to supplement and update several reviews published recently [17,18,19].

## 2. Nisin A

Discovered in 1928 [20] nisin A is the prototypical and best-studied lantibiotic, secreted by some *Lactococcus lactis* strains. Low toxicity and safety of use in food contributed to the worldwide success of nisin as a natural food preservative (E234), especially in dairy and canned products [21,22,23]. Although nisin is a complex polycyclic, positively charged, molecule consisting of 34 amino acids (molecular weight 3500 Da) it is quite readily available by fermentation, so is relatively cheap.

The remarkable commercial success of this compound has positioned it at the pinnacle of bacteriocin research. These studies are concentrated not only on knowledge attained with respect to genetics, structure, mode of action and chemical properties but also for the potential of its practical applications. It is an FDA-approved, GRAS (generally regarded as safe) peptide with recognized potential for possible clinical use [24]. Additionally, nisin’s immunomodulatory properties, analogous to those described for many human defense peptides, suggest its possible role as a novel immunomodulatory therapeutic [25].

Nisin is built from five lanthione bridges which help impart a conformation that is key to its antibacterial properties. Unusual amino acids present in this antibiotic are dehydroalanine, dehydrobutyrine and β-methyl-lanthionine (Figure 1) [26]. Dehydroamino acids are formed via post-translational processing of a ribosomally synthesized precursor polypeptide. 

It is an antibiotic of dual action, which is a consequence of the presence of two distinct structural domains located at the N- and C-termini of the peptide (Figure 2) [27,28,29]. The N-terminal fragment of a peptide docks to lipid II with high affinity via interaction with lipid II pyrophosphate group, finally forming a cage-like structure by intermolecular hydrogen bonds.

At the same time, the C-terminus of nisin flips into the membrane and enables the formation of pores, thus causing cell leakage, namely immediate efflux of nutrients and small compounds, resulting in cell death (Figure 2). The flexible, short hinge region connects the N- and C- terminal regions and is crucial for C-terminal fragment translocation through cell membrane and pore-forming activity.

The activity of nisin against Gram-negative bacteria is much lower than that against Gram-positive bacteria, mainly because lipid II is located at the inner membrane, and the rather impermeable outer membrane in Gram-negative bacteria prevents nisin from reaching lipid II [30].

## 3. Design of Novel Antibacterials by Modification of Nisin Structure

The structural features and mode of action make nisin suitable for its structure engineering. These modifications are performed in order to generate nisin variants with enhanced antimicrobial activity, better solubility, and ability to evade resistance. Enhancement of activity against Gram-negative bacteria is one of the major driving forces of these studies. Thus, the obtained results should allow progress to be made in overcoming the inherent shortcomings of nisin. The two major means that have been used for this purpose are bioengineering and chemical modifications of the structure of this antibiotic.

### 3.1. Bioengeeneering by Mutagenesis

Due to the complex nature of nisin’s structure, chemical synthesis of its analogues is challenging, especially in economic terms. This leaves a huge space for the efforts of bioengineering, especially if considering possible industrial preparation of the analogs by fermentation. Recent years have seen a growing number of studies on the use of genetic tools (engineering within the cell) and the use of synthetic biology-based (in vitro engineering) approaches [17,18,19,31], mostly in order to advance the understanding of the fundamentals of the structure–antibacterial activity relationship. 

Bioengineering could be performed either by applying genome mining, design and preparation of hybrid-peptides (by coupling lanthipeptides with variable antibacterial peptides) and especially by utilization of mutation techniques. As a result, the number of identified nisin analogs have increased tremendously in the last decades [19].

There are at least five naturally occurring variants of nisin A, which differ by up to 10 amino acids [32]. The most variable ones are produced by *Streptococcus agalactiae* and *Straptococcus uberis*. Because of their gene-encoded nature they could easily undergo bioengineering procedures, which resulted in obtaining libraries of analogs suitable for elucidating structure–function relationships [30]. Nisin variants are usually produced by transfection of *Escherichia coli* with the nisin biosynthesis pathways or the expression of nisin variant genes in a wild type nisin producing strains, from which the nisin gene has been deleted.

The simplest examples are mutations of the hinge region, in which flexibility is crucial for C-terminal membrane translocation. Additionally, the three residues, which form this region, affect antibiotic stability and solubility. The first approach was concentrated on studying the replacement of one of its three amino acid (Asn, Met or Lys) by randomly introduced proteinogenic ones using site-saturation mutagenesis. These studies initially brought quite promising results, providing variants which displayed enhanced activity against Gram-negative bacteria [33]. The generation of nisin derivatives with enhanced activity against Gram-positive pathogens was achieved later, in 2009, using a non-targeted approach. In this instance, the use of a random mutagenesis-based approach created over 8000 derivatives among which only one variant displayed an activity higher than the parent nisin against a human pathogen—*Streptococcus agalactiae* [34].

Mutations, consisting of changes in the length of hinge region, did not dramatically affect nisin production in host bacteria [33]. Deletion of one amino acid or insertion of three additional ones usually led to a sharp decrease in antimicrobial activity [35]. Analogs of the hinge region consisting of five or six amino acids usually exhibited moderate improvement of antimicrobial effects [36]. These results show that the hinge region might still be considered as a potential target for bioengineering. 

The high activity of nisin also results from formation of stable pores in bacterial membranes. Pores are formed by the C-terminal fragment of this antibiotic (amino acids 23–34, rings D and E, see Figure 1) and are believed to cause rapid dissipation of transmembrane electrostatic potential, which in turn results in membrane permeabilization and rapid bacterial cell death. Although quite limited, studies on bioengineering of this fragment provided some valuable variants of improved solubility at neutral pH [37], resistance to the specific immunity system present in some bacteria [38] or of increased antibacterial activity [39,40,41].

The N-terminal fragment of nisin (amino acids 1–12) binds to the bacterial cell wall precursor—lipid II. The two lanthionine rings A and B (Figure 1) form a pyrophosphate cage around its headgroup. Binding of this fragment of nisin to lipid II induces the formation of large aggregates in the membranes of bacteria. Consequently, this part of the molecule is most important for antibacterial activity and thus it is not surprising that significant effort was devoted to its modification. 

Early engineering attempts were targeted at the N-terminal ring A (amino acids 3–7). Amino acids in positions 4, 5 and 6, namely in a region that is at the border of the pyrophosphate cage [25], suggest large mutational freedom available and thus represent suitable targets for mutagenesis. Indeed, some of the variants displayed improved activity against several non-pathogenic indicator strains [42]. Notably, several natural nisin variants as well as novel nisin-like peptides (agalacticin, flavucin, moraviensicin and maddinglicin) possess a lysine at position 4. Its presence ensures the interaction of positively charged lysine side chain with negatively charged phosphate and indeed brought a positive effect [43]. Ring B of nisin was found to be far less amenable to amino acid substitution [42,44]. For example, analysis of 144 variants in which position 10 (glycine) was targeted revealed that the majority of the obtained clones either exhibited activity comparable to the mother nisin or appeared to be completely inactive [44]. Another lysine (amino acid 12) is located between rings B and C. The site-saturation approach applied for replacement of this amino acid led to the identification of a small number of variants with improved antibacterial activity, however these studies did not derive any structure–activity relationship [45]. 

Advanced, highly sensitive solid-state NMR study on binding of nisin to lipid II present in a model membrane consisting of mixture of artificial phospholipids brought identification of Ile4, Lys12, Ser 29 and the whole hinge region as functional hotspots that are critical for the cellular adaptability of nisin [46]. Indeed, in a series of double and triple mutants those derived from mutations of at least two of these hotspots exerted the highest antibacterial activity [40,42,47,48]. One of them displayed promising activity against *Listeria monocytogenes* [48], a Gram-positive strain that causes listeriosis, an invasive disease affecting pregnant women, neonates, the elderly, and immunocompromised individuals.

Although nisin remains the only lantibiotic that is so extensively modified by application of mutagenesis, the full use of any variant as a therapeutic entity has not yet been fulfilled. The progress of mutagenetic studies, although quite slow and cumbersome, is clear. However, the successful clinical development of nisin analogs requires more detailed understanding of the mechanism of nisin pharmacology and creative improvement of the discovered modification procedures. Many knowledge gaps are still waiting for closure.

### 3.2. Variants Containing Non-Canonical Amino Acids

Logic extension of the studies described above was introduction of nonproteinogenic (non-canonical) amino acids into discrete positions of nisin. Generally, there are two complementary approaches for the incorporation of non-canonical amino acids into the peptide chain of interest. The first is residue-specific and exploits the translational machinery of the host, which accepts the structural analog of the amino acid as a surrogate of a classic one. The second is site-specific, which is far more complex and requires the additional presence an orthogonal aminoacyl-tRNA-synthetase-tRNA pair, which do not cross-react with the endogenous expression machinery. This orthogonal pair uses a stop codon as a coding one to incorporate a chosen non-canonical amino acid into the protein of interest. Thus, the first step to biosynthetic incorporation of nonproteinogenic amino acids into lantipeptides was the elaboration of novel molecular biology tools [49,50,51,52,53].

Analogs of proline and methionine have been incorporated into nisin at positions of their parent amino acids without gene manipulation (Figure 3A), by feeding nutritionally deficient strains with these analogs [53,54,55,56]. In addition, variants containing tryptophan, obtained previously by mutagenesis, were used as substrates for introduction of tryptophan-like residues (Figure 3B) [57]. Although analogues of improved antibacterial activity were obtained in some cases, these substitutions did not bring spectacular effects. It is worth noting, however, that the major goal of these mutations was their further chemical modification by using reactive groups of the introduced analogs (see Section 4.1). 

A site-specific procedure has been used for modification of the structure of nisin with analogs of phenylalanine [52,53] and derivatives of lysine (Figure 4) [52,58]. Interesting results gave application of chemically reactive (*N*-chloroacetylamino)phenylalanine, which replaced cysteine in position 3 of the nisin precursor and thus disturbed the formation of the ring A. Spontaneous reactions of the chloroacetyl fragment with nucleophilic amino acids of this mutant provided several analogs with novel macrocyclic ring topologies. Unfortunately, none of the variants exhibited promising activity.

### 3.3. Hybrid Molecules

Combining of nisin, or its fragments, with other antimicrobial peptides is performed on the premise of receiving potent and novel antimicrobial drug candidates of multiple modes of action. So far, only single examples of the use of bioengineering approach for that purpose have been described. 

The outer membrane of Gram-negative bacteria constitutes an efficient protective barrier that prevents nisin from reaching the cellular membrane and then exerting its antimicrobial action. This inconvenience was addressed by fusion of nisin, or its *N*-terminal fragments, with short antimicrobial peptides that combine different functionalities. These include peptides whose designs are based on statistical analyses and natural antimicrobials produced by various organisms (crocodile, honeybee, scorpion, frogs and penis fish). The obtained hybrids are able to pass the outer membrane of Gram-negative organisms while retaining as much nisin antimicrobial function as possible at the cytoplasmic membrane, which is documented by their up to 12-fold activity if compared with nisin [59,60,61].

The biosynthesis system of the nisin was used to obtain the set of hybrids containing the N-terminal lipid II binding motif of nisin (amino acids 1–22) and the distinct small C-terminal lipid II binding motifs derived from other lantibiotics—haloduracin or lacticin [62]. One of the variants, termed TL19, exerted quite promising antimicrobial activity against *Enterococcus faecium*, a bacterial strain used as probiotics, but also known to produce various complicated infections (abdominal, skin, urinary tract, and blood). In a similar study the same fragment of nisin was conjugated with variants of small insect defense peptides yielding a series of hybrids [63]. The variants of insect thanatin (produced by *Podisus maculiventris*) and rip-thanatin (produced by *Riptortus pedestris*) were obtained by replacing their disulfide bridges with just a single sulfur. Some of these novel antibiotics (see the representative example in Figure 5) showed substantial antimicrobial activity against Gram-negative pathogens. Moreover, they appeared to be highly resistant against the nisin-resistant proteins—nisinase and specific protease.

## 4. Chemical Modifications of Nisin

Chemical modifications have attracted far less interest and are quite scarce. Due to the complex nature of lanthipeptide structures, chemical synthetic strategies are challenging. This is well documented by a total synthesis of nisin and mutacin rings A and B performed by solid state peptide synthesis. The study was carried out in order to understand the important structural factors underlying highly selective molecular recognition between these fragments of the antibiotics and lipid II by using NMR [64]. The conjunction of these rings, though, might lead to novel nisin variants that are far too expensive.

### 4.1. Click Reaction as a Tool for Nisin Modification

Most of the described procedures of nisin modification consider the use of nisin variants carrying acetylenic or azide moieties and are thus suitable for application as substrates in a click reaction [65]. The first attempt was to conjugate an N-terminal fragment of nisin (amino acids 1–12) functionalized with propargylamine with vancomycin derivatives bearing 2-amino-3-azidopropane at C-terminus (Figure 6). Their coupling via a microwave version of the click reaction provided hybrids of significantly higher activity than the starting components [66]. 

In a similar manner nisin analogues composed of its 22-amino-acid, *N*-terminal fragment and of its C-terminal fragment mimics, in which ring structure is built up by the replacement of lanthionines by double bonds (Figure 7A), have been obtained. In this case C-terminal asparagine of the ABC fragment was used in a form of 3-azidopropylamide while the mimic contained an alkyne moiety. The new products obtained had similar activity to the wild-type nisin [67]. The strong binding of these two classes of variants to lipid II suggests that modification of the C-terminal fragment of nisin does not deteriorate the biological activity.

It is worth to note that these functionalizations are possible thanks to digestion of nisin with trypsin and chymotrypsin, which selectively generate nisin fragments composed of, correspondingly, rings AB and ABC. 

Connecting the ABC fragment of nisin with hydrophobic poly(octahydroindole-2-carboxylic acid), containing additional lysines at the C-termini, provided another interesting class of nisin mimics [68]. Lysines were introduced because they exist in cationic forms at physiological pH and thus improve solubility of the obtained variants (Figure 7B). 

In addition, nisin AB fragment was conjugated with variable linear peptidoids, which are known to have increased in vivo stability compared to the corresponding peptides [69]. These hybrid peptide—petidoids (representative structure shown in Figure 7C) were shown to have low micromolar activity (i.e., comparable to natural nisin) against methicillin-resistant Staphylococcus aureus. Similar synthetic procedure was applied to the preparation of a big series of nisin mimic, in which its C-terminal fragment is replaced by a series of cationic peptides and peptoids with known antibacterial action and pore-forming properties [70]. Hybrid peptides, where a hydrophilic PEG4 linker was used, showed good antibacterial activity against *Micrococcus luteus*. One of these peptides, a hybrid of nisin ABC fragment and a 22-amino-acid antibiotic derived from magainin—pexiganan—is shown in Figure 7D. 

Functionalization of nisin with fluorescent reporter molecules was designed for studies on the mode of action of this antibiotic. They were obtained from nisin C-terminally substituted with propargylamine and a series of functionalized fluorescent probe azides (for representative structure, see Figure 8). Results of the studies show that these nisin derivatives retain both their antimicrobial activity and their membrane permeabilizing properties [71]. Additionally, they indicate that the attachment of a relatively large molecular entity to the C-terminus of nisin means it can be readily transported through bacterial membranes. This finding might encourage the functionalization of nisin C-termini with low-molecular weight antibiotics (for example, a cycloserine) in order to smuggle them through the bacterial membrane. 

In addition, the possibility of the residue-specific incorporation of fluorescent dyes into certain fragments of nisin was demonstrated. This was performed by using nisin analogs containing either azidohomoalanine or homopropargylglycine available from the bioengineering approach (see Section 3.2). This resulted in a library of variants bearing fluorescent dyes at various positions of the nisin peptide chain (the representative example is shown in Figure 8) [72]. All the resulting conjugates retained antimicrobial activity, which substantiates the potential of this method as a tool to further study nisin localization inside the cell and of its molecular mode of action.

Lipopeptides are nontoxic, biodegradable, highly stable, ecofriendly, and nonpolluting biomolecules [73]. These properties make them good candidates for use as antimicrobial agents in medicine and food science. In order to find out an efficient method for the synthesis of nisin–lipid conjugates (nisin lipopetides), click chemistry was selected as the method of choice. Coupling of 1-undecyne (model lipid) with nisin variants containing diaza-alanine in variable chain positions proved useful in this procedure [56]. This procedure was successfully used to couple the C-terminal AB fragment of nisin with a series of long chain alkynes [74]. The obtained lipopeptides exhibited quite promising antibacterial activities.

### 4.2. Late-Stage Functionalization of Nisin

Functionalization of intact nisin and its variants by chemical introduction of specific functionalities in one step has not yet received suitable attention. For direct functionalization there are available: C-terminal carboxylic moiety, four free amino groups (N-terminal one and three from side chains of lysines), and three dehydroamino acids. In addition, the possibility of modifications of two side chains of methionine could be considered (Figure 9), though there is a lack of such studies. The number of reports on direct modifications of nisin or its fragments is also limited. 

Especially surprising is the practical lack of the modifications of nisin by the Michael reaction. This reaction is quite commonly used for the modification of peptides and proteins containing dehydroamino acids [75] and results from the reactivity of the double bonds of the unsaturated amino acids. The oldest report considers acid-catalyzed addition of a water molecule to both dehydroalanines of nisin [76]. The resulting derivatives (amido-alcohols) appeared to be unstable and yielded peptidyl fragments of low activity. This reaction seems to explain the low stability of nisin in aqueous solutions. In addition, glutathione adds non-specifically to the double bonds of the two dehydroalanines of nisin, which renders the nisin molecule inactive [77].

A palladium-mediated cross-coupling reaction of nisin with a series of phenylboronic acids gave mixtures of the Heck reaction (dehydrophenylalanines are formed) and conjugated addition to double bonds (phenylalanines are produced) (Figure 10). The reaction is quite complex, but could be steered to some extent by appropriate manipulation of reaction conditions; however, the Heck reaction always predominates [78]. All the three dehydroamino acids react, yielding mixtures of mono-, di- and three-substituted products (for a total of six products), with the C-terminal dehydroalanine reacting as the first one. It has to be emphasized that, despite of formation of complex mixtures of products, this late-stage modification approach is far more efficient than the alternatives.

Reactivity of the other groups of nisin, except for synthesis of substrates using the click reaction, are scarcely used for functionalization of this antibiotic. Thus, a variety of hydrophobic amines were coupled with the C-terminal AB fragment to generate semisynthetic lipopeptides that display potent inhibition of bacterial growth [74]. A representative example shown in Figure 11 exhibited quite potent activity against vancomycin-resistant Enterococci. Experiments with membrane models indicate that these semisynthetic constructs operated via a lipid II-mediated mode of action without causing pore formation.

There are a number of reports of structural and chemical modification of proteins with reducing sugars, following the Maillard reaction, which is demonstrated by changes such as fluorescence and browning. The Maillard reaction is associated with the development of UV-absorbing intermediate compounds, prior to generation of brown pigments. These intermediates are characterized by a high proportion of carbonyl groups that can react with the amino moieties of proteins. Thus, Maillard products induced by gamma irradiation (40 kGy) of solutions of glucose or dextran reacted with nisin—providing mixtures of novel conjugates with extended activity towards Gram-negative bacteria along with improved antioxidant activity [79]. This makes them interesting candidates for food protection.

Free amino groups of nisin were also used for its functionalization with gellan by formation of amides with its carboxylic moieties [80]. The reaction was catalyzed by two popular peptide-coupling agents active in aqueous solutions. Gellan is an edible exopolysaccharide produced by *Sphingomonas paucimobilis* possessing many advantages such as biocompatibility and stability against most microbes, enzymes, and many chemicals. The idea on its use was to protect nisin against deactivation; the antibacterial duration of nisin against *Staphylococcus epidermidis* increased 2–4 times in comparison with the mother antibiotic, lasting for several days.

## 5. Conclusions

Bacteriocins can be considered unconventional antimicrobial molecules which have attracted great interest because they possess many of the attributes essential for the treatment of infections caused by multi-drug-resistant bacteria and to be alternatives to traditional antibiotics. Nisin is the most commercially important member of this group and is approved as a food protectant in over 50 countries. Being readily available and cheap, it is also the most widely studied and exploited. However, its clinical potential and possible use in human and animal therapy has not yet been fulfilled, in part due to its low solubility and stability in body fluids—thus the intensive research focused on enhancement of its functionality in terms of specific activity, spectrum of activity, solubility and/or temperature and pH stability. 

The broad range of technologies that has been developed for the engineering of nisin has resulted in expanding the knowledge of the factors vital for nisin’s mode of action and provided many variable tools for producing and modifying lantibiotic peptides, as well as enabling numerous nisin variants of variable structures to be obtained. However, nisin is quite a complex molecule and it is not surprising that these studies do not allow us to draw meaningful relation between nisin analog structures and their antibacterial activity. Thus, further multifaceted studies are required.

The studies also reinforced the belief that bioengineered lantibiotics can contribute to a solution of antibiotic resistance across a broad range of bacterial pathogens. The most promising seem to be the results on the introduction of non-canonical amino acids into the structure of this antibiotic, as well as the studies on conjugates of nisin (or nisin fragments) with variable antibacterials. These studies, however, are in their infancy.

## Figures and Tables

**Figure 1 pharmaceuticals-16-01058-f001:**
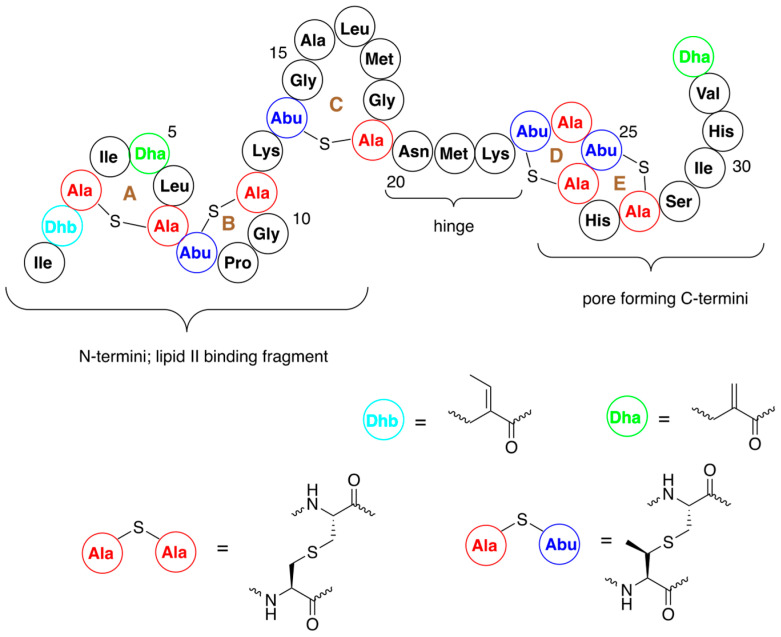
Structure of nisin A with numbering of amino acids and indication of A–E rings. Structures of dehydroalanine (Dha) and dehydrobutyrine (Dhb) and lanthionine and β-methyl-lanthionine are shown.

**Figure 2 pharmaceuticals-16-01058-f002:**
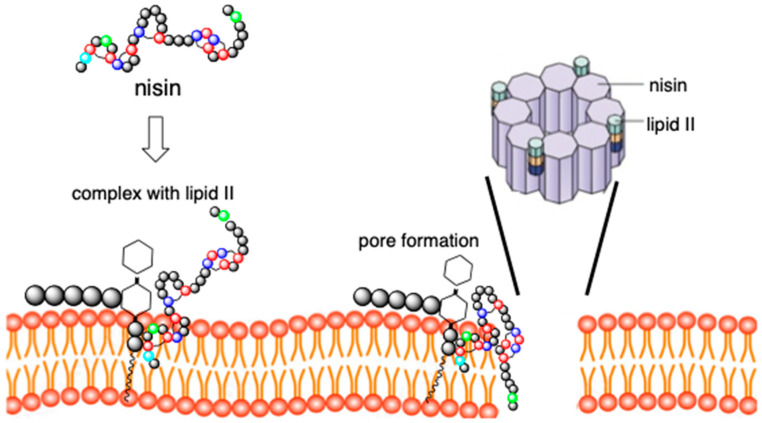
Dual mechanism of action of nisin. Nisin binds to lipid II and then undergoes conformational change forming pore in the membrane. Pore is composed of eight C-terminal fragments of nisin.

**Figure 3 pharmaceuticals-16-01058-f003:**
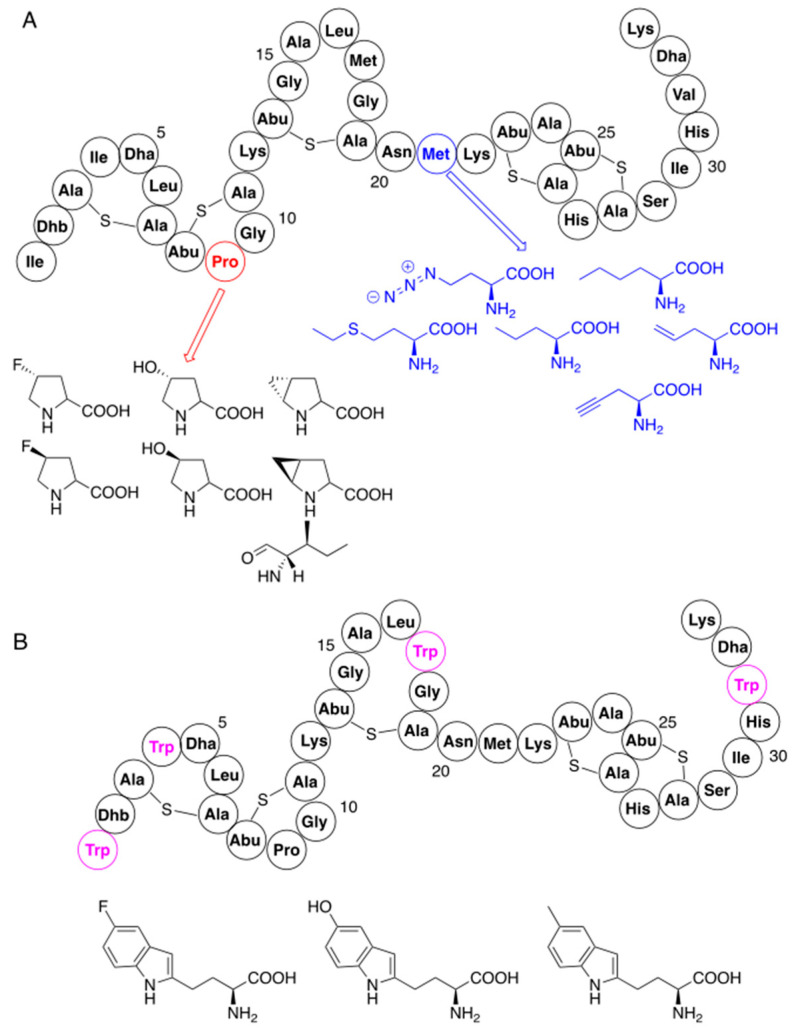
Residue-specific modifications of nisin with non-canonical amino acids: (**A**) substitutions of lysine and methionine; (**B**) modifications with tryptophan analogs using previously mutated variants (nisin mutation sites with tryptophan are marked in magenta). Structures of non-canonical amino acids are also shown.

**Figure 4 pharmaceuticals-16-01058-f004:**
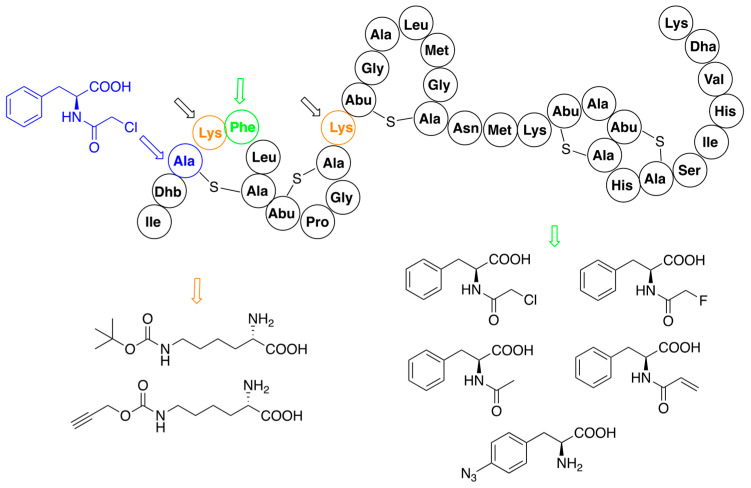
Residue-specific modifications of nisin variants containing lysine (black arrows, mutation sites shown in orange) and phenylalanine (red arrows, mutation site shown in green). Site of replacement by (N-chloroacetylamino)phenylalanine (blue arrow) is shown separately. Structures of introduced non-canonical amino acids are indicated.

**Figure 5 pharmaceuticals-16-01058-f005:**
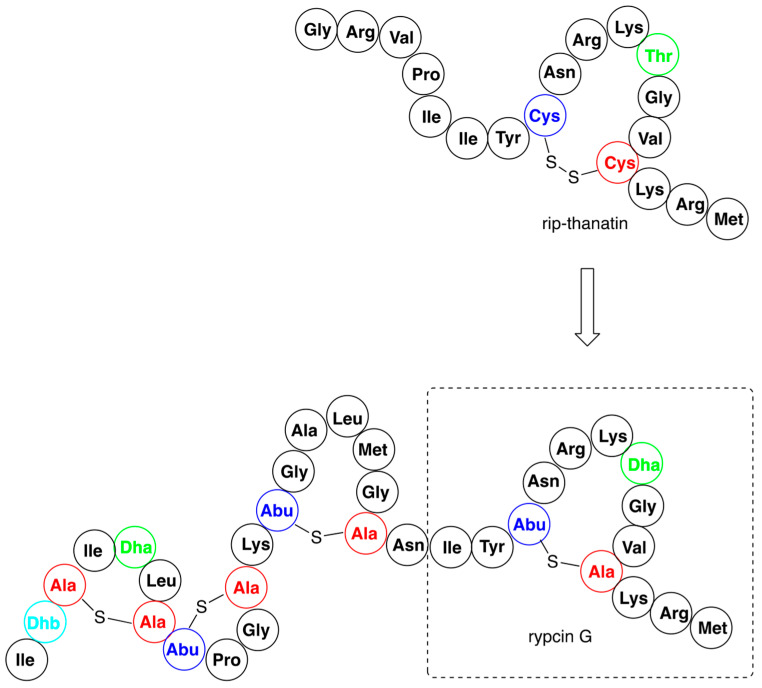
An example of antibacterial nisin hybrid—rypcin G, alongside with the structure of parent rip-thanatin.

**Figure 6 pharmaceuticals-16-01058-f006:**
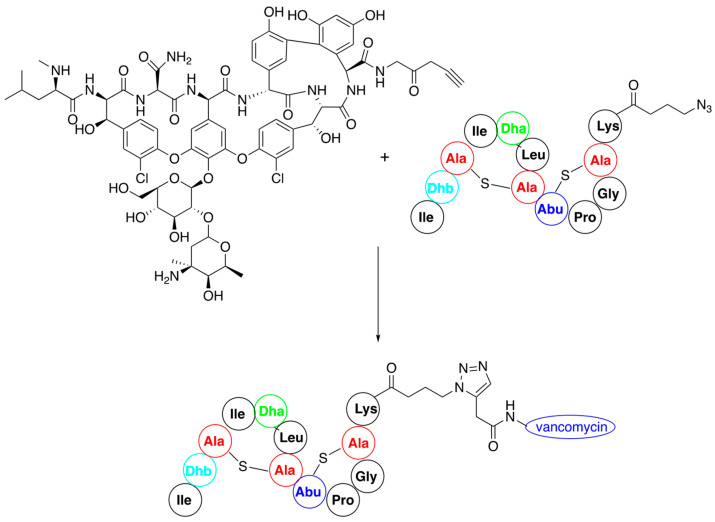
Synthesis of nisin–vancomycin conjugate by click reaction of their derivatives.

**Figure 7 pharmaceuticals-16-01058-f007:**
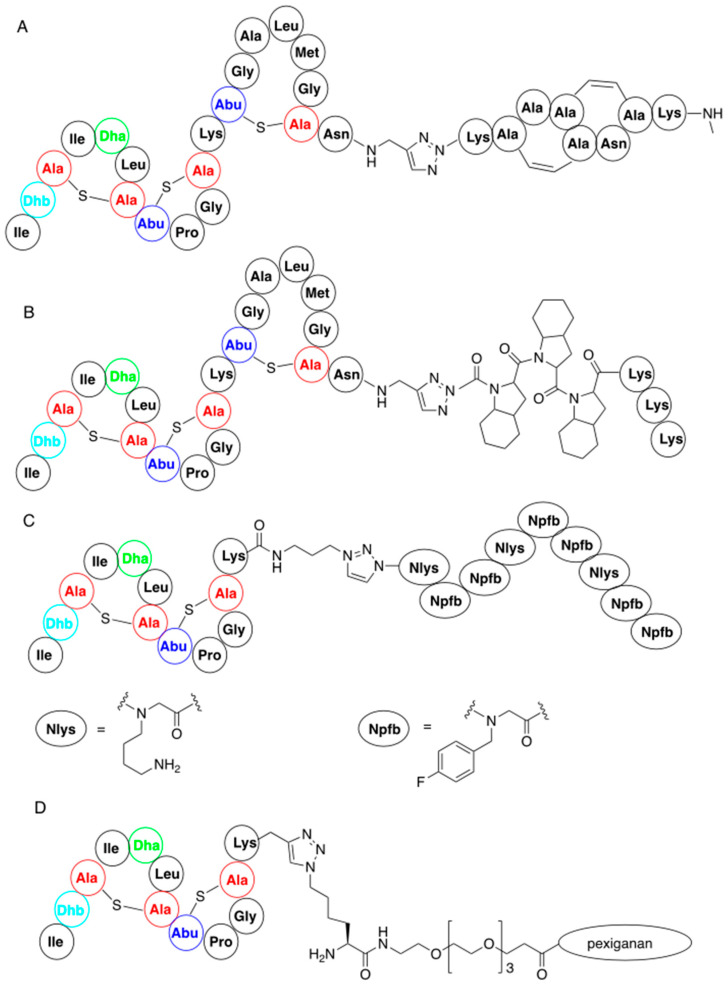
Representative nisin-derived conjugates: (**A**) variant of nisin containing artificial C-termini; (**B**) variant containing C-terminal analog of polyproline; (**D**) nisin–polypeptidoid conjugate; (**C**) conjugate of nisin N-termini with pexiganan.

**Figure 8 pharmaceuticals-16-01058-f008:**
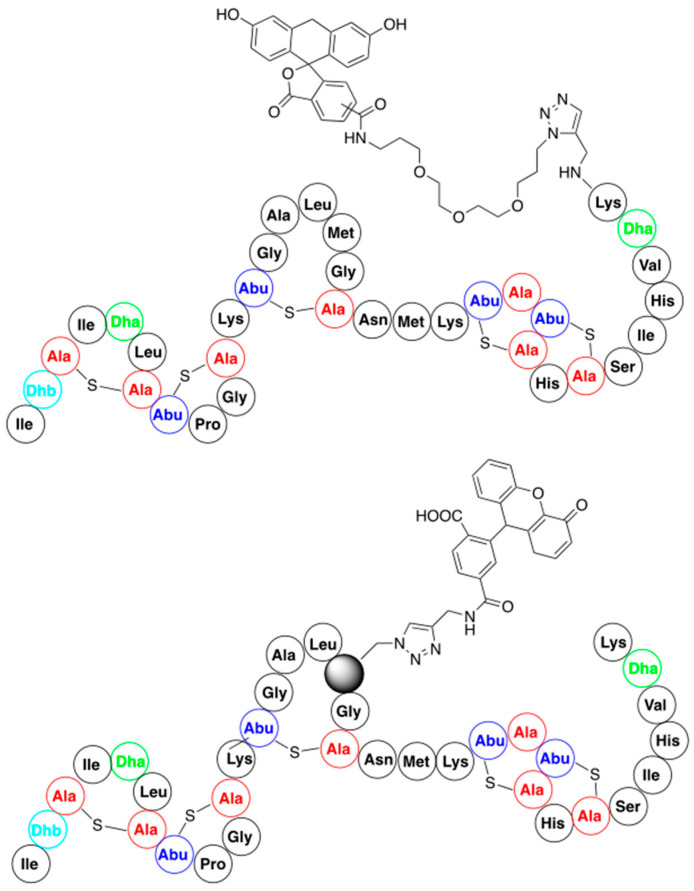
Structures of representative nisin-fluorescent reporter conjugates.

**Figure 9 pharmaceuticals-16-01058-f009:**
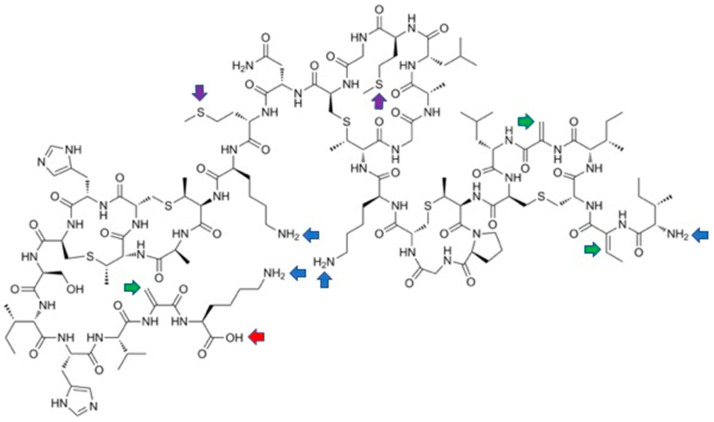
Possible sites for last-stage modifications of nisin showing reactive entities: blue arrows—free amino groups; red arrow—carboxylate; green arrows—dehydroamino acids; purple arrows—methionines.

**Figure 10 pharmaceuticals-16-01058-f010:**
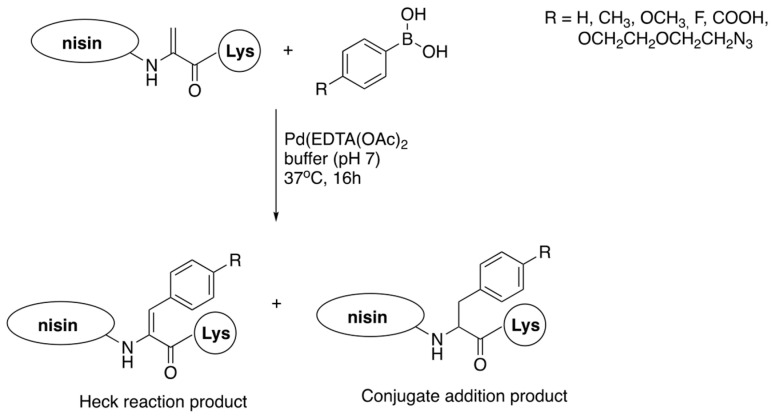
Reaction of nisin C-terminal dehydroalanine with phenylboronic acids.

**Figure 11 pharmaceuticals-16-01058-f011:**
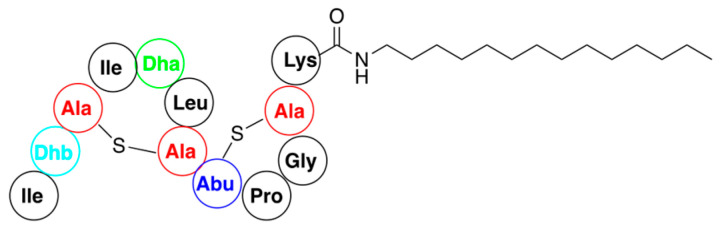
Representative structure of nisin lipopeptide obtained by direct acylation of amines with AB fragment of antibiotic.

## Data Availability

Not applicable.

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
