# Peer review of "Engineering of Nisin as a Means for Improvement of Its Pharmacological Properties: A Review"

_pharmaceuticals, 2023, doi:10.3390/ph16081058_

Round 1
Reviewer 1 Report
The authors tried to review employed modifications on nisin through bioengineering and chemical modifications. The manuscript is interesting in this regard; however, they should inform us, compared to similar articles, what is the novelty of their review. The abstract should name various approaches applied for nisin modification and the conclusion section should briefly compare the most prevalent or successful approaches. The conclusion section is not informative and needs to be completely rewritten to enlighten briefly on how we can enhance the antimicrobial activity, antibacterial spectrum, and physicochemical properties (solubility, solution stability and protease resistance) of nisin. There are some concerns regarding the bibliography mentioned as follows:
●Authors should define the name of databases, the range of years, and the keywords that were used to find literature. There are some articles related to this topic that do not appear in the reference list such as:
Li Q, Montalban-Lopez M, Kuipers OP. Increasing the antimicrobial activity of nisin-based lantibiotics against Gram-negative pathogens. Applied and environmental microbiology. 2018 Jun 15;84(12):e00052-18.
●The second and third sentences of the abstract are redundant. They should be revised.
●The Genus and species should be in italics such as Streptococcus agalactiae and Straptococcus uberis on page 3 line 110. These are only two examples.
●Sections 3-1, 3-2, and 3-3 require extensive English editing
●The quality of Figure 2 should be increased.
●The abstract and conclusion need complete revision as mentioned in the general comments.
Extensive to moderate English modifications are required.
Author Response
Please see the attachemnet

Reviewer 2 Report
Comments to authors:
Point 1: The abstract is poorly written; a summary of the work would be mentioned in the abstract including the physiochemical characteristics of the Nisin andas well as more details of the Lantibiotics.
Point 2: The authors should rewrite the title again to make it more informative and suitable for the content.
Point 3: Authors would clarify the aim of the manuscript at the end of introduction.
Point 4: Figure 2. is very pixelated, please reconsider.
Point 5: Could you add graphic abstract to the manuscript.
Point 6: Figure 3. is repeated and not adding much to the readers.
Point 7: you mention that Nisin has potential for clinical use, please clarify and mention examples.
Also,
1- Please add the list of abbreviations and follow the instructions of authors.
2- Add a prospective future to the conclusion.
3- The language of the whole manuscript should be revised.
The language of the whole manuscript should be revised.
Author Response
Please see tha attachement

Reviewer 3 Report
The manuscript entitled "Modifications of Nisin as a Mean for the Development of Novel Antibacterials" by Musiejuk and Kafarski review the engineering efforts of nisin, in order to improve its antimicrobial activity and physicochemical properties (solubility, solution stability and protease resistance). The manuscript is fairly organized, but the overal scientific quality is low and many of the examples provided are poorly described, in some cases without references. The English language usage is poor and a thorough revision of the manuscript is also recommeded.
The names of species should be written in italics, there are several examples throughout the manuscript.
Figure legends are poor and do not help understanding the pictures.
Other issues:
lines 10-11: delete "limit their availability for clinical studies and further pharmaceutical commercialization. "
line 50: to obtain modified lantibiotic....
line 70: "Other unusual amino acids present in this antibiotic are ..." Why "other" if none was mentioned before?
line 134: Deletion of one amino acid or...
line 144: The following sentence needs further explanation: "...resistance to specific resistance system present in some bacteria"
line 151: correct the typos: "the moleculle is vital for antibacteril activity..."
line 160: correct the typos: "of positively charde lysine with...
line 162: amino acid
line 164: For example, analysis of...
lines 173-174: the sentence should not be a new paragraph.
line 175 : what is the meaning of "a very interesting variant of promising activity"?
line 179: desribed
line 188: re-write: "First one is residue-specific one,"
line 192: A dot is missing
line 202: Paragraph 4.1?
line 310-311: Re-write: Second report considers the use of C-terminal AB ring for coupling with a series of long chain alkynes.
lines 361-368: at least a reference should be provided.
line 374: This is no way to constrcuct a sentence: "Being readily available and cheap it is also the best studied and exploited one."
The manuscript entitled "Modifications of Nisin as a Mean for the Development of Novel Antibacterials" by Musiejuk and Kafarski is written in poor English and a thorough revision of the manuscript is required.
Author Response
Please see tha attachement

Round 2
Reviewer 1 Report
Modifications are acceptable.
Author Response
Reviewer: Modifications are acceptable.
Response: We would like to thaks once more for careful reports.
Reviewer 3 Report
The manuscript pharmaceuticals-2491056, entitled "Modifications of Nisin as a Mean for the Development of Novel Antibacterials", by Kafarski and Musiejuk was revised taking into consideration some of the criticisms raised and it significntly improved its quality. Although some descriptions continue to be superficial without scientific details and reading like a advertisement, its global quality increased. There are still some issues related to th eEnglish usage. I recommend a thorough revision of the manuscript focusing on the grammar usage.
Other details: when refering to the molecular mass, indicate the units (line 67).
line 79: Correct "...acids and are formed..."
line 214: "see paragraph 4.1". Better call it section, subsection, not paragraph.
The manuscript was revised and its quality increased. I still think thatthe manuscript contains some language issues.
Author Response
Review: he manuscript pharmaceuticals-2491056, entitled "Modifications of Nisin as a Mean for the Development of Novel Antibacterials", by Kafarski and Musiejuk was revised taking into consideration some of the criticisms raised and it significntly improved its quality.
Replay: We would like once more to thank the Reviewer for effort and detailed reports.
Review: Although some descriptions continue to be superficial without scientific details and reading like a advertisement, its global quality increased.
Replay: Maybe our belief that modifications of nisin are indeed tha way to obtain better antimicrobials made that the paper remins advertisement a little bit. Of course, we do know that other means are possible - for example tteh use of suitable carriers (there are single paper on this subject published).
Review: There are still some issues related to th eEnglish usage. I recommend a thorough revision of the manuscript focusing on the grammar usage. & I still think that the manuscript contains some language issues.
Replay: We have additionally corrected English (marked in magenta in the text)
Round 3
Reviewer 3 Report
The manuscript was revised and all the criticisms raise dto the previous version were solved. There ar eonly a few minor issues that require the action of authors:
line 127: "banks of analogs". Instead, use "libraries of analogs"
line 130: .. of nisin variant genes
line 329: terminus
The English usage has improved.
Author Response
The manuscript was revised and all the criticisms raise dto the previous version were solved. There ar eonly a few minor issues that require the action of authors:
line 127: "banks of analogs". Instead, use "libraries of analogs"
line 130: .. of nisin variant genes
line 329: terminus
We would like to thank Reviewer once more. All the errors have been corrected (now marked in gray)